# Dual-Stream Feature Extraction Network Based on CNN and Transformer for Building Extraction

**Liegang Xia** [1,*] **, Shulin Mi** [1] **, Junxia Zhang** [1] **, Jiancheng Luo** [2] **, Zhanfeng Shen** [2] **and Yubin Cheng** [1]

1   College of Computer Science and Technology, Zhejiang University of Technology, Hangzhou 310023, China
2   Institute of Remote Sensing and Digital Earth, Chinese Academy of Sciences, Beijing 100875, China
*   Correspondence: xialg@zjut.edu.cn; Tel.: +86-571-8529-0027

**Abstract:** Automatically extracting 2D buildings from high-resolution remote sensing images is among the most popular research directions in the area of remote sensing information extraction. Semantic segmentation based on a CNN or transformer has greatly improved building extraction accuracy. A CNN is good at local feature extraction, but its ability to acquire global features is poor, which can lead to incorrect and missed detection of buildings. The advantage of transformer models lies in their global receptive field, but they do not perform well in extracting local features, resulting in poor local detail for building extraction. We propose a CNN-based and transformer-based dual-stream feature extraction network (DSFENet) in this paper, for accurate building extraction. In the encoder, convolution extracts the local features for buildings, and the transformer realizes the global representation of the buildings. The effective combination of local and global features greatly enhances the network's feature extraction ability. We validated the capability of DSFENet on the Google Image dataset and the ISPRS Vaihingen dataset. DSEFNet achieved the best accuracy performance compared to other state-of-the-art models.

**Keywords:** building extraction; convolutional neural network (CNN); high-resolution remote sensing image; transformer; semantic segmentation

## 1. Introduction

Automatic extraction and analysis of 2D building features from high-resolution remote sensing images is a popular research direction [1]. The research results from this area have been widely used in many fields, such as the detection of changes [2,3], city and country planning, and assessment of natural disasters [4], etc. However, remote sensing images with a high resolution have complex backgrounds with redundant features, making it difficult to extract buildings [5]. Due to the fast growth of deep learning, the analysis and utilization of remote sensing images at high resolution has been made more effective [6]. Compared with human visual interpretation, deep learning models can actively learn potential semantic building information and achieve automatic extraction, which demonstrates their great advantages and application potential in the field of remote sensing [6].

Compared with natural scene images, it is more challenging to segment 2D buildings in high-resolution remote sensing images. There are three main reasons for this: (1) Differences in the architectural style and form of buildings (such as size, materials, color, shape etc.), which cause multiscale problems in buildings [7]. (2) Buildings are widely distributed and backgrounds are complex and diverse, which can cause false detection and missed detection [7]. (3) The proportion of building foreground is much smaller than that of natural images, which leads to the unbalanced category problem in neural network learning [8]. In addition, due to tree and shadow occlusion [9], viewing angle [10], etc., it is difficult to estimate buildings accurately. Therefore, how to effectively use the rich spatial, spectral, and semantic information in remote sensing images with a high resolution to accurately acquire building information is among the most challenging tasks in the field of remote sensing.

Semantic segmentation has powerful feature extraction and representation capabilities, which is conducive to accurate two-dimensional building extraction and positioning. A CNN uses a downsampling operation, which effectively expands the network's receptive field, while acquiring multiscale features, which improves the global context information modeling ability. However, a CNN with excessive depth will cause the gradient to disappear, and the details of small objects will be severely reduced or even completely lost. This indicates that the CNN receptive field is limited. Limited by its receptive field, a CNN is good at local feature extraction but cannot capture global features well for global context modeling [11]. With the development of transformer [12] technology and its powerful self-attention global context modeling capability, transformer technology has achieved a performance beyond CNNs in many basic tasks in the computer vision field. A transformer effectively models global contextual information [13], but it does not perform well in extracting local detailed features, resulting in erroneous predictions of local details. To considering the spatial relationship and detailed characteristics of the building, it is necessary to combine global and local information.

Moreover, the downsampling operations of semantic segmentation reduce image resolution and detailed information gradually disappears, resulting in a blurred building boundary prediction [3,14]. Semantic segmentation is a prediction task at the pixel level, its application in building segmentation tasks requires the assistance of accurate edge information. The use of shallow edge information can compensate for the lack of building details caused by downsampling operations and drive the network to focus more on building edge information, thus improving the network's performance.

We propose a CNN-based and transformer-based dual-stream feature extraction network (DSFENet). In the encoder, a CNN and transformer simultaneously perform feature extraction and feature aggregation at each scale stage, to achieve the combination of local features and global features, and to achieve multiscale features. Meanwhile, we designed a semantic embedding module and a spatial embedding module, to address the edge blurring problem in semantic segmentation.

The primary contributions of this paper are the following:

1. We proposed a CNN-based and transformer-based dual-stream feature extraction network. The CNN branch enriches detailed features in the transformer branch, and the transformer branch enhances global features in the CNN branch, realizing a complementary representation of local features and global features.
2. We designed a semantic embedding module and a spatial embedding module, to enable the fusion of low and high level features, to achieve a combination of edge details and semantic information, in order to improve the semantic segmentation results of buildings.
3. Our model achieved a higher performance than current state-of-the-art networks on both the Google Beijing dataset and the ISPRS Vaihingen dataset.

## 2. Related Work

The building segmentation task is among the key tasks in the remote sensing field. CNN is currently the main technique used for building semantic segmentation. Meanwhile, the transformer model has given more potential to semantic segmentation. Some pure transformer networks, as well as combinations of CNN and transformer structures, have been proposed.

### 2.1. CNN-Based Semantic Segmentation for Building Extraction

CNNs are a mainstream technology, and significant progress has been achieved. DeepResUnet [15] effectively performs pixel-level segmentation of urban buildings and produces high-quality segmentation results. The cascaded downsampling part is for the extraction of the building feature map, then the upsampling part is for the reconstruction of the extracted feature map. A spatial residual inception [16] network is used to capture and gather multiscale context through sequential fusion of features at multiple levels. It is hard

to recognize buildings of various scales using a single receptive field. In order to efficiently capture and restore features, AGBEDNet [17] includes an attention module based on a grid and an atrous spatial pyramid pooling module. SSPDNet [18] was proposed for accurate building segmentation, utilizing an augmented encoder and two-stage decoder consisting of selective spatial pyramid expansion modules for better extraction and recovery of critical multiscale information. MAP-Net [19] extracts the preserved local spatial multiscale features through several parallel modules. Each module serves to extract corresponding high-level features with a specific resolution of semantics. The above studies show that the integration of multiscale features is very important in the task of building extraction.

### 2.2. Transformer-Based Semantic Segmentation for Building Extraction

In the past two years, transformer technology has achieved significant progress in semantic segmentation, and some transformer-based models have obtained a performance better than CNNs. SETR [20], SegFormer [21], and Swin-UNet [22] deployed pure transformer models for semantic segmentation. With the global context modeled in every layer with self-attention, these models provide powerful semantic segmentation capabilities. The pure transformer model applied to natural or medical images demonstrated its powerful capabilities and provides new technical support for building extraction.

There are two main problems with transformer-based architectures: a high computational load, and low edge classification accuracy [23]. The efficient transformer in [23], based on the Swin transformer [24], proposed an enhanced edge method to deal with the problem of low object edge accuracy for semantic segmentation. Sparse token transformers (STT) [25] represent buildings as a collection of "sparse" feature vectors, by introducing a sparse token sampler. STT learns the long-term dependency of tokens, to achieve accurate building extraction. Currently, there are fewer building extraction methods for pure transformer models.

### 2.3. CNN and Transformer-Based Semantic Segmentation for Building Extraction

The inductive bias in the CNN architecture renders it lacking of a representation of long-range dependencies in images [11]. Architectures based on a transformer encode global features using a self-attention mechanism. The combination of CNN and transformer architectures may be an effective method to improve semantic segmentation performance. A medical transformer [26] was developed as the gated axial-attention model. This extends the module of self-attention by adding a control method. In order to obtain multiscale long-range dependencies, UTNet [27] applies self-attentive modules in both encoding and decoding processes. TransUNet [28] combines a transformer with U-Net, where the transformer transforms tokenized image patches from CNN feature maps into input sequence encodings for global context extraction. UNetFormer [29] uses the lightweight ResNet18 as the encoder, combined with the decoder of the transformer architecture, to realize the extraction of global-local features, with a lighter structure. ST-UNet [30] uses a parallel dual-encoder structure of a Swin-transformer and CNN. The two encoders are executed separately and combine the output of the Swin-transformer encoding block into the CNN encoding block at each layer, to realize the parallelization and fusion of the dual encoders. Zhang [31] et al. proposed a transformer and convolutional neural network (CNN) hybrid deep neural network. The network is based on a Swin transformer as the encoding structure and CNN as the decoding structure. At the same time, it uses the auxiliary boundary detection branch to constrain the boundary of the segmentation result, to obtain finer segmentation results.

For building extraction, the spatial position relationship of the building is important, and the local details of the building also need to be guaranteed. The fusion architecture of a CNN and transformer can realize a complementarity between the global features and local features of buildings. The efficient hybrid transformer (EHT) [32] has a structure of an encoder based on a CNN and a decoder based on a transformer. The STransFuse [33] model combines a transformer with a CNN, extracting feature representations at multiple

semantic scales and adaptively fusing semantic information between features with a self-attentive mechanism. MSST-Net [34] uses a Swin transformer as the base structure of the encoder. Moreover, this network individually decodes the feature maps at different levels and uses convolutional fusion. The above models demonstrate the possibility of fusing CNN and transformer architectures.

## 3. Materials and Methods

Accurately extracting buildings was the main goal of this paper. The complementary relationship between local features and global representation was the focus of our study. We designed DSFENet based on a dual-stream feature extraction encoder, aiming to enhance feature representation learning using the local features of convolution operation and the global representation of a self-attention mechanism. The feature aggregation module (FAM) was aimed at aggregating local and global features across different scale stages. It enhances the feature representation capability of the encoder. Meanwhile, a semantic embedding module (SEEM) and spatial embedded module (SPEM) were implemented. This achieved the fusion and complementation of edge features at low level and semantic features at high level, to obtain building polygons with accurate edges. Figure 1 shows the structure of the network.

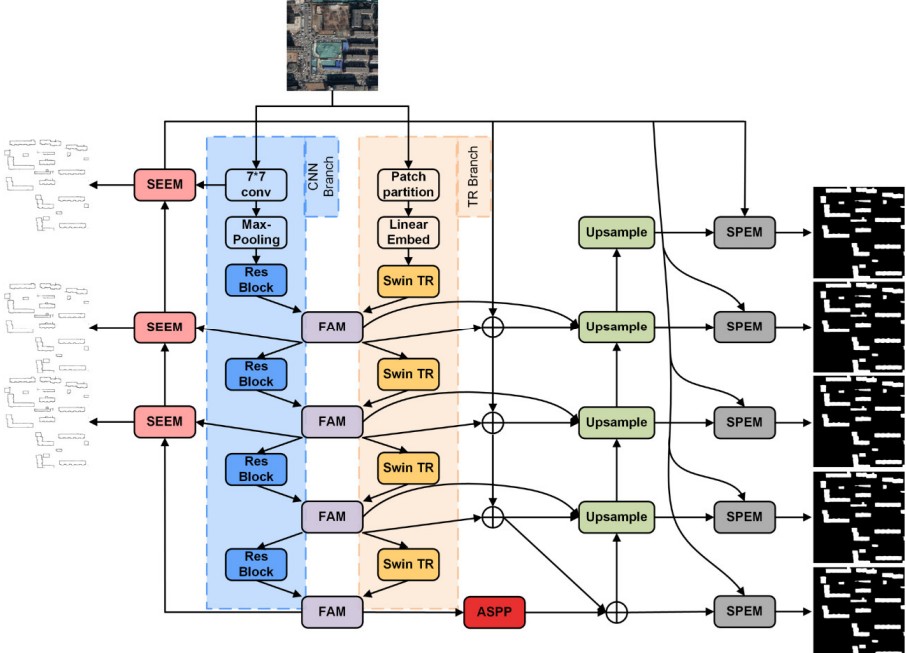

**Figure 1.** The architecture of DSFENet. Res Block represents residual block. Swin TR represents Swin transformer. FAM is the feature aggregation module. SEEM and SPEM are the semantic embedding module and spatial embedding module, respectively. Upsample is an upsampling module consisting of a bilinear interpolation and a convolution operation. ASPP is an atrous spatial pyramid pooling module.

### 3.1. Dual-Stream Feature Extraction

DSFENet uses a dual-stream feature extraction encoder as its core structure. The encoder composed of a CNN branch based on ResNet34 [35] and a transformer branch based on a Swin transformer [24]. The local features of the CNN branch make up for the shortage of partial details of the transformer branch and improve the detail performance of building prediction. The transformer branch's global features enhance the CNN branch's contextual information modeling effect, effectively reducing false detections and missed detections in the building extraction.

### 3.1.1. CNN Branch

The CNN branch follows the ResNet34 design based on residual blocks. Each res-block includes two successive 3 × 3 convolutions and a short skip connection. The CNN performs five downsampling operations to build a feature pyramid. The channel count of the feature map at each stage grows with the depth of the network, and the resolution reduces accordingly. The number of residual blocks in each stage is 2, 3, 4, and 3. During the convolution operation, the CNN branch can continuously provide local feature details to the transformer branch. Figure 2 illustrates the construction of the residual block.

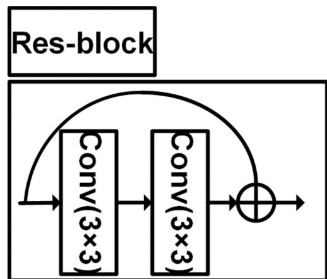

**Figure 2.** Res block.

### 3.1.2. Swin Transformer Branch

The transformer branch is based on the design of the Swin transformer block. It applies a patch merge layer responsible for downsampling and increasing the channels of serialized data, to achieve multiscale feature representation. The Transformer branch is downsampled four times to correspond to the CNN branch multiscale feature. The number of Swin Transformer blocks at each stage is 2, 2, 2, and 2. The number of multihead attentions at each stage is 4, 8, 16, and 32. The self-attention principle is to perform attention interaction between the window and the shifted window, realizes global context modeling, and provides global information for the CNN branch. Figure 3 shows the Swin-transformer block.

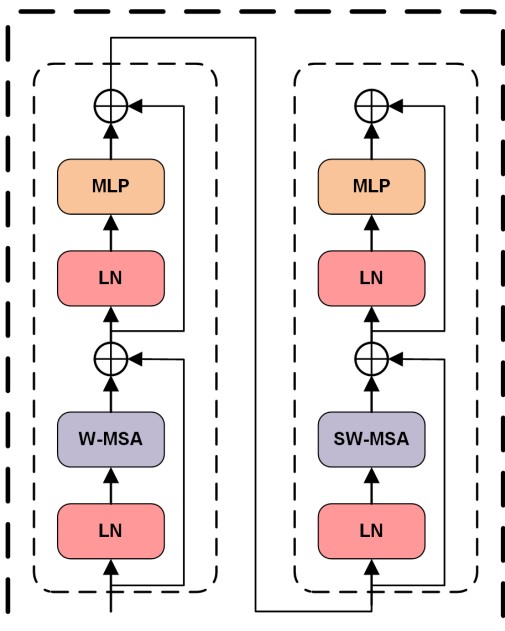

**Figure 3.** Swin-transformer block.

### 3.2. Feature Aggregation Module

As the transformer features are serialized, unlike CNN features, the feature aggregation module aims to build a bridge for feature fusion. First, the transformer's one-

dimensional serialized features are transformed into traditional multidimensional image features using the transformer's position encoding. Using positional encoding for feature transformation can avoid excessive transformation loss. Then, the concatenation operation is performed with the CNN features in the channel dimension. The aggregated features are then processed separately, one using $1 \times 1$ convolution to reduce the dimensionality of the feature and restore it to its original size, then input to the CNN branch. The other part encodes the position of the aggregated features, and then concatenates them into one-dimensional serialized features for input to the transformer branch. Since the CNN and transformer branches capture different levels of features (local features and global features), a feature aggregation module is used to eliminate the semantic differences between them and enhance the local detail awareness and global modeling capabilities. Figure 4 shows the feature aggregation module.

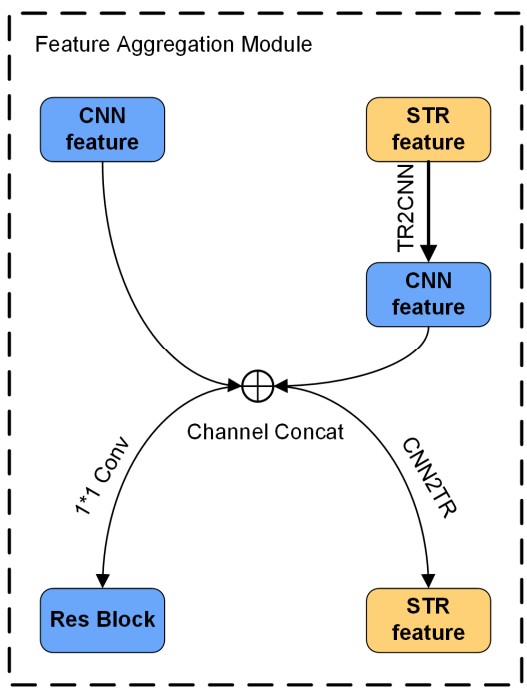

**Figure 4.** Feature aggregation module. STR feature represents the feature of the Swin transformer. TR2CNN, the serialized data is spliced into multidimensional features. The fused features are processed separately. The number of feature channels is recovered using $1 \times 1$ convolution, then it is input to the CNN branch. CNN2TR function is used to map the multidimensional features, convert them into serialized features, and input them into the transformer branch.

### 3.3. Semantic Embedding Module and Spatial Embedding Module

Shallow networks are more sensitive to low-level information, while deep networks are more sensitive to high-level information, such as semantic features. Currently, semantic segmentation focuses more on multiscale feature fusion or expanding the receptive field, while ignoring the importance of low-level features such as edge information [3]. It is not feasible to simply add low-level and high-level features or channel concatenating operations. Cluttered low-level information may introduce noise, cause interference, and reduce semantic information accuracy. Therefore, we designed a semantic embedding module (SEEM) and spatial embedding module (SPEM). The SEEM uses the semantic information to perform an attention operation on messy shallow information, to reduce the shallow information redundancy. Detailed low-level information is fused with multiscale high-level semantic features with the SPEM, to enrich the feature details. Features at different scales contain different semantic information, and spatial information extracted at different levels can be fused to highlight detection targets.

### 3.3.1. Semantic Embedding Module

The shallow layer of the network is more responsive to the features of the object, including color information and edge information. However, we observed the features generated by the shallow information and found that the shallow information is often redundant and contains much messy information. Although most of the structural information of buildings can be correctly learned and predicted, a large number of redundant non-building features are also revealed. It is hard to separate useful building information from information that cannot be used. The deeper the network, the more sensitive it is to the accuracy of the semantic information of the target. Therefore, using deep semantic information to add an attention mechanism to the edge information allows the neural network to focus on the edge with correct semantic information about the shallow information. This approach reduces distracting redundant information and improves the building details.

This module accepts low-resolution semantic features and high-resolution edge features. It first uses convolution operations to perform channel alignment operations on semantic features and edge features, and then uses bilinear interpolation operations to restore low-resolution semantic features to the high-resolution edge feature size. It uses the Sigmoid function to activate the semantic features. The purpose is to obtain a global attention map and use the attention map and edge features for a dot product operation to highlight semantically correct edges and remove redundant edge information and noise. Finally, it uses a convolution action to recover the channel count of the feature map. The module is shown in Figure 5.

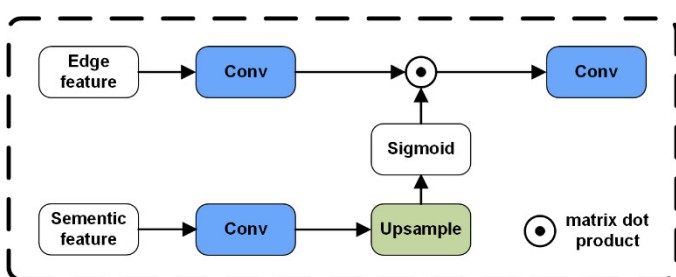

**Figure 5.** Semantic embedding module.

### 3.3.2. Spatial Embedding Module

This module accepts edge and semantic features as input. The edge features at this stage become clear after the attention operation of the semantic embedding module, which reduces the redundant information and noise. First, it uses a convolution operation to align the semantic and edge features, then it uses a bilinear interpolation operation to restore low-resolution semantic features to the high-resolution edge feature size. Then, two pixel-level feature maps of identical size are added. The purpose is to use the semantic edge features to recover the edge-detail features lost from the semantic features. Finally, the module uses convolution to recover the channel count of the feature map. Figure 6 shows the specific architecture of the module.

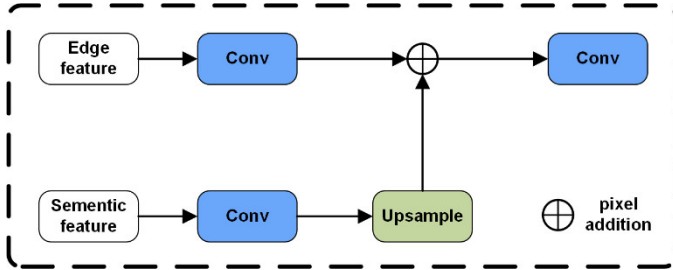

**Figure 6.** Spatial embedding module.

### 3.4. Pyramid Skip Connection

The richer features are fused at the decoder stage, which is an efficient way to recover lost features. Some models such as UNet3+ [36] design a special structure to reuse features between the encoder and decoder, to improve accuracy. However, many feature connections and too much reuse can lead to further computing resource shortages. To further alleviate the computing resource shortage, we reconsidered the feature connection structure between the encoder and decoder. A feature pyramid connection structure [37] was used, which can easily and quickly perform multiscale feature fusion. High-level semantic features at each scale are combined with downsampled high-resolution low-level details, which are then passed to the decoder at the same height. The skip connection is used to deliver the original semantic information to the decoder of the same height. Therefore, each decoder layer contains features from the encoder, semantic features are fused with edge information, and these feature maps can effectively capture fine-grained details. The feature connection not only greatly reduces the redundancy of feature reuse but also improves the building extraction accuracy.

### 3.5. Loss

#### 3.5.1. Polygon Loss

We use binary cross entropy $Loss_{bce}$ and Dice coefficient loss for polygon supervision. The BCE loss is shown in Equation (1).

$$Loss_{bce} = -\sum Y_i \log(\widehat{Y}_j) - \sum (1 - Y_i)\log(1 - \widehat{Y}_j) \tag{1}$$

Calculating the similarity among samples usually employs the Dice coefficient. The Dice loss formula is shown in Equation (2):

$$Loss_{dice} = 1 - 2 \times \frac{\hat{Y} \cap Y}{\hat{Y} + Y} \tag{2}$$

$\hat{Y}$ and $Y$ represent prediction results and labels, respectively, and $\hat{Y} \cap Y$ is the intersection of and $\hat{Y}$ and $Y$.

The polygon loss is shown in Equation (2), which is the sum of the two losses:

$$Loss_{polygon} = Loss_{bce} + Loss_{dice} \tag{3}$$

#### 3.5.2. Semantic Edge Loss

Only 10% of the ground truth is edge and 90% is non-edge, due to severe deviations in the edge and non-edge pixel distribution. We used the class-balanced (CMSE) loss function [35]. Specifically, dynamic weights were used to the equalize edge and non-edge pixels. Equation (4) shows the CMSE loss formula:

$$Loss_{cmse} = \frac{1}{m}\sum \left(\beta\big((\hat{Y}_j \in |Y-|) - Y_j\big)^2 + (1 - \beta)\Big((\hat{Y}_j \in |Y+|) - Y_j\Big)^2\right) \tag{4}$$

where $|Y-|$ stands for edge pixels and $|Y+|$ stands for non-edge pixels, respectively. $\hat{Y}$ is the predicted polygon result, subscripted $j \in [0, 1, \dots H \times W]$. H represents the label height, and W represents the label width. The CMSE loss uses class balance weight $\beta$ to counteract the disequilibrium between the edge and non-edge. $\beta$ is calculated dynamically using the following formula:

$$\beta = \frac{Sum_{edge}}{Sum_{edge} + Sum_{nonedge}} \tag{5}$$

$Sum_{edge}$ is the amount of edge pixels on the edge label. $Sum_{nonedge}$ is the amount of non-edge pixels on the edge label. $\beta$ is the non-edge pixel weight, and $1 - \beta$ is the edge pixel weight.

## 4. Experiments and Results

### 4.1. Datasets

We selected two high-resolution remote sensing image datasets to evaluate DSFENet: Google Image and Aerial Image.

The first is the Beijing dataset, which contains different types of buildings. Most of the buildings are traditional style buildings or modern style buildings. The traditional style buildings are mostly royal buildings and residential buildings in the form of low-rise courtyards. The modern buildings are mostly uniform high-rise residential buildings and modern high-rise buildings. We chose a 0.536 m resolution 18-level Google image.

The second dataset is the ISPRS Vaihingen dataset released in 2018. This dataset is an airborne image dataset from Vaihingen, Germany. The overall number of buildings in the dataset is large, and the buildings are relatively dense. These complex scenes require high algorithm robustness. The remote sensing image format of the dataset is an 8-bit TIFF file with a resolution of 9 cm. The images consist of three bands: green, near-infrared, and red.

Images from the Beijing and Vaihingen datasets were cropped to $512 \times 512$ pixels, because they were too large to fit in memory. For the Beijing dataset, we manually drew and generated building labels. The Vaihingen dataset is a multiclass dataset; therefore, we extracted building type annotations to generate building labels and utilized a contour finding algorithm to generate edge labels. The datasets were split according to the ratio of 80% for the training set and 20% for the test set. The Beijing dataset had 590 images for training and 148 images for testing. The Vaihingen dataset had 677 images for training and 170 images for testing. A sample of the Beijing and Vaihingen datasets is shown in Figure 7.

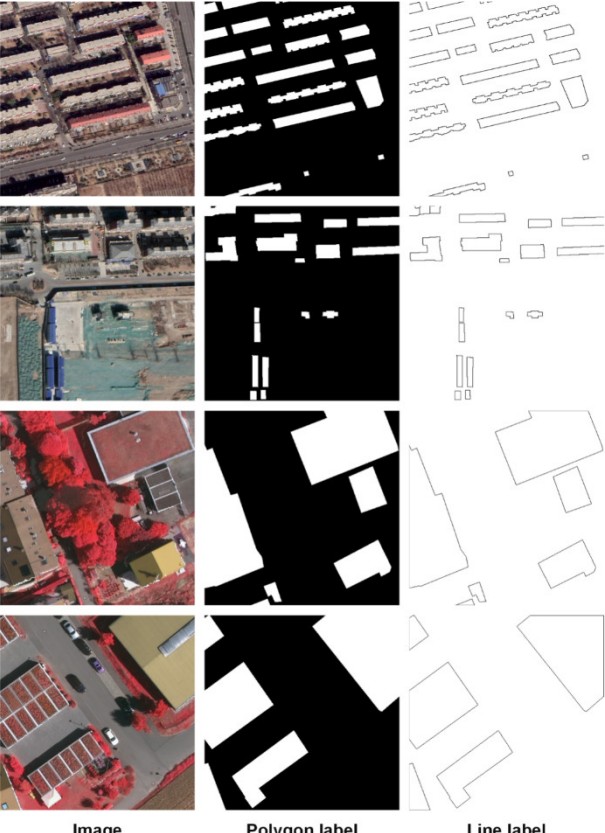

**Figure 7.** Sample of the Beijing and Vaihingen datasets. The first two rows are examples in the Beijing dataset. The last two rows are examples in the Vaihingen dataset.

### 4.2. Implementation Details

The networks in our experiments are based on PyTorch 1.7 and CUDA 11.0 framework implementations. DSFENet and all comparison methods were trained on an NVIDIA RTX 3090 graphics card with 24G memory. We used the ImageNet pretrained weights of the ResNet34 model to initialize the weights of the DSFENet CNN branch. The batch size of both datasets was 4. The learning rate was initially $2 \times 10^{-4}$. The learning rate was updated to 1/4 of the total epoch. A total of 400 epochs were trained for both datasets. We also used data augmentation operations including rotation, translation, and horizontal flip.

### 4.3. Evaluation Metrics

The performance of the methods was evaluated using the F1 score, intersection over Union (IoU), and boundary IoU [38]. The IoU metric can reflect the accuracy of building segmentation. The boundary IoU method can reflect the accuracy of the building edge. The reason for using the expansion kernel to expand the edge is because the single-pixel edge ratio is low, there may be a slight offset, and the direct calculation of the boundary IoU may have low accuracy, which cannot truly reflect the accuracy of the edge. Considering that a slight offset is acceptable, this study expanded the edge pixels to reflect the accuracy of the edge more realistically. This method used a kernel size = 5 pixels to expand the edge, and then calculated the accuracy using Equation (6). Exp is expansion, ks is kernel size, A is the prediction result, and B is label.

$$IoU = \frac{A \cap B}{A \cup B} \tag{6}$$

$$Boundary\ IoU = \frac{\exp(A, ks) \cap \exp(B, ks)}{\exp(A, ks) \cup \exp(B, ks)} \tag{7}$$

The F1 score is able to evaluate binary classification models. Before calculating the F1 score, some metrics had to be calculated. In the formula below, $C_{tp}$ represents the amount of positive pixels of correctly recognized buildings. $C_{tn}$ represents the amount of negative pixels correctly recognized as non-buildings. $C_{tp}$ represents the amount of non-building negative pixels misidentified as building-positive pixels. $C_{tn}$ represents the number of positive pixels of buildings misidentified as negative pixels of non-buildings. Precision is the proportion of true positive pixels among all positive pixels.

$$Precision = \frac{C_{tp}}{C_{tp} + C_{tn}} \tag{8}$$

Recall is the proportion of pixels correctly identified as positive.

$$Recall = \frac{C_{tp}}{C_{tp} + C_{fn}} \tag{9}$$

Equation (10) shows the F1 score formula, where $P$ is precision and $R$ is recall.

$$F1\ score = 2 \times \frac{P \times R}{P + R} \tag{10}$$

### 4.4. Results

We compared our model with state-of-the-art models. The models were classified into the following types: (1) CNN-based networks, including U-Net [39], D-LinkNet [40], and UNet3+ [36]; (2) multitask learning networks, including EGNet [41] and DDLNet [22]; (3) transformer-based networks, including Swin-UNet [22]; and (4) networks based on a CNN and transformer, including MedT [26], TransUNet [28], and UTNet [27].

On the Beijing dataset, DSFENet achieved a boundary IoU of 0.5334, an IoU of 0.7752, and an F1 Score of 0.8752. DSFENet obtained the highest performance for these three indicators, surpassing the performance of the other models. U-Net, U-Net3+, and D-

LinkNet achieved 0.7015, 0.7120, and 0.7194 for the IoU indicators. DSFENet's performance greatly surpassed them, which shows the effectiveness of the transformer architecture. EGNet, based on multitask learning, only achieved a boundary IoU of 0.3012 and an IoU index of 0.6371, which was not as good as U-Net. DDLNet achieved relatively satisfactory results, with a boundary IoU of 0.5321 and IoU of 0.7662. Swin-UNet generated fuzzy and disordered building predictions, resulting in the lowest accuracy for all three indicators. The reason for this result may have been that the size of the Beijing dataset was too small to allow the neural network of the pure transformer architecture to converge. MedT, TransUNet, and UTNet achieved accuracy results close to D-LinkNet, and none of them achieved satisfactory results. Table 1 and Figure 8 show the DSFENet and other models on the Beijing dataset.

**Table 1.** The results of semantic edge detection on the Beijing dataset.

| Study Area | Method | Boundary IoU | IoU | F1 Score |
|---|---|---|---|---|
| Beijing | U-Net | 0.4599 | 0.7015 | 0.8267 |
| | U-Net3+ | 0.3908 | 0.7120 | 0.8413 |
| | D-LinkNet | 0.4640 | 0.7194 | 0.8433 |
| | DDLNet | 0.5321 | 0.7662 | 0.8703 |
| | EGNet | 0.3012 | 0.6371 | 0.7887 |
| | Swin-Unet | 0.2883 | 0.5234 | 0.6976 |
| | MedT | 0.3544 | 0.5975 | 0.7545 |
| | TransUnet | 0.4452 | 0.6948 | 0.8257 |
| | UTNet | 0.4427 | 0.6799 | 0.8176 |
| | DSFENet | **0.5334** | **0.7752** | **0.8752** |

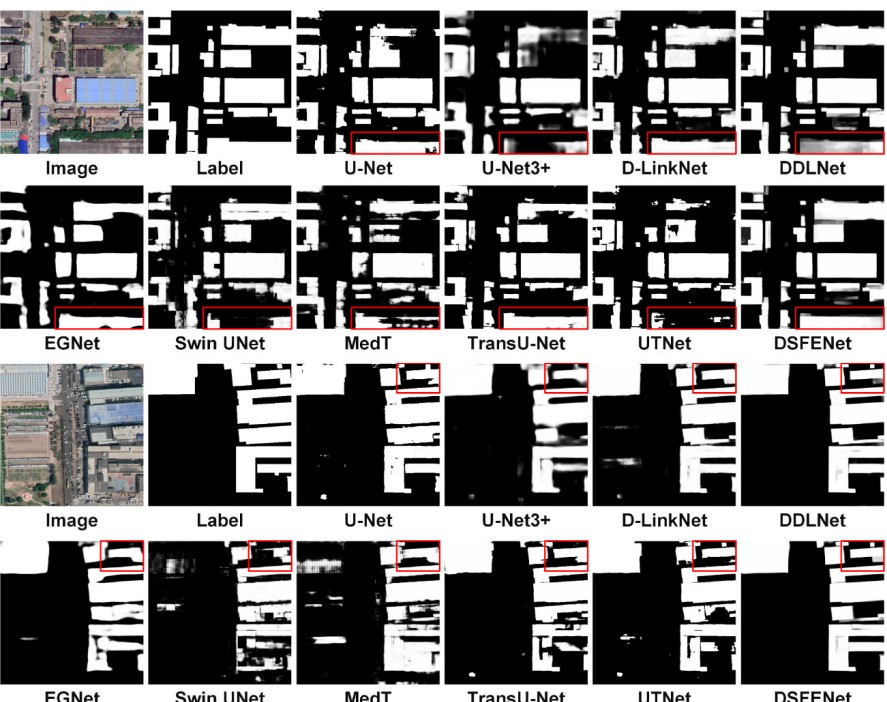

**Figure 8.** DSFENet comparison experiment results on the Beijng dataset.

Compared with the Beijing dataset of Google images, aerial images have a higher resolution and richer details. The performance of most models is also better. DSFENet achieved a boundary IoU of 0.5706, an IoU of 0.9294, and an F1 Score of 0.9637. DSFENet obtained the highest performance for these three indicators. U-Net, U-Net3+, and D-LinkNet achieved 0.8892, 0.8969, and 0.9279 for IoU indicators respectively. EGNet and DDLNet achieved results with IoU values of 0.8871 and 0.9280, respectively. Swin-UNet

only achieved an IoU of 0.7440. MedT, TransUNet, and UTNet had 0.7737, 0.8921, and 0.8992, respectively, for the IoU index. Table 2 and Figure 9 show the DSFENet and other models with the Vaihingen dataset.

**Table 2.** The results of semantic edge detection on the Vaihingen dataset.

| Study Area | Method | Boundary IoU | IoU | F1 Score |
|:---:|:---:|:---:|:---:|:---:|
| | U-Net | 0.4758 | 0.8892 | 0.9405 |
| | U-Net3+ | 0.4235 | 0.8969 | 0.9459 |
| | D-LinkNet | 0.5370 | 0.9279 | 0.9628 |
| | DDLNet | 0.5432 | 0.9280 | 0.9629 |
| | EGNet | 0.4025 | 0.8871 | 0.9405 |
| Vaihingen | Swin-Unet | 0.2224 | 0.7440 | 0.8492 |
| | MedT | 0.2614 | 0.7737 | 0.8666 |
| | TransUnet | 0.4696 | 0.8921 | 0.9470 |
| | UTNet | 0.4641 | 0.8992 | 0.9421 |
| | DSFENet | **0.5706** | **0.9294** | **0.9637** |

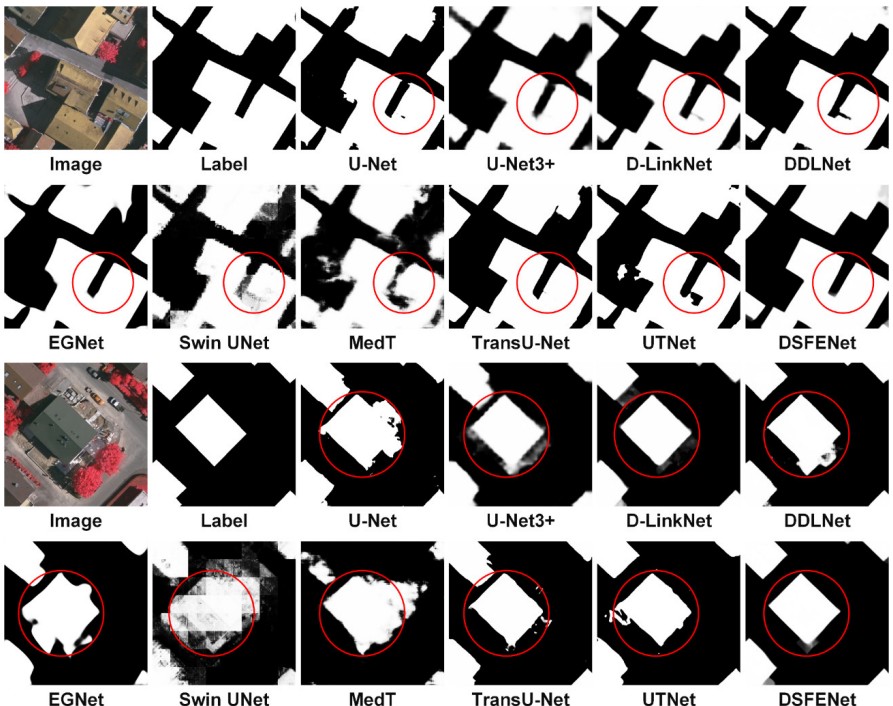

**Figure 9.** Comparative experimental results of DSFENet on the Vaihingen dataset.

From the experimental and experimental results, it can be seen that DSFENet provided a better detection precision and visual effect in comparison with the other models. This proved that the integration of local and global features worked. TransUNet, MedT, and UTNet all failed to achieve satisfactory outcomes. This might have resulted from the dataset size and the method of combining CNN features with transformer features.

### 4.5. Ablation Study

This section used ablation experiments to evaluate the impact of the CNN branch and transformer branch on the overall results of the DSFENet. In this section, the encoder part was redesigned according to DSFENET, and DSFENet-CNN with only the CNN branch and DSFENet-TR with only the transformer branch were tested. Table 3 shows the experimental results.

**Table 3.** Accuracy comparison of ablation experiments.

| Study Area | Method | Boundary IoU | IoU | F1 Score |
|---|---|---|---|---|
| Beijing | DSFENet | **0.5334** | **0.7752** | **0.8752** |
| | DSFENet-CNN | 0.5253 | 0.7739 | 0.8695 |
| | DSFENet-TR | 0.4746 | 0.7219 | 0.8388 |

Compared with DSFENet-CNN and DSFENet-TR, DSFENet with a dual-stream branch achieved the best accuracy performance. This demonstrates the importance of the dual-stream feature extraction branch. DSFENet-TR did not perform as well as DSFENet-CNN. Neural networks based on a transformer architecture require large amounts of training data to realize better performance [42]. Due to the size limitations of the Beijing dataset, it could not achieve better accuracy results. Due to the complementarity of global information and local information, the structure of the two-stream branch effectively alleviated the large amount of data required by the transformer architecture, accelerated network convergence, and achieved a higher accuracy. DSFENet-CNN achieved accuracy results close to DSFENet, which also shows that the CNN performs better in extracting local information. A detailed comparison of the ablation experiments is shown in Figure 10. It can be seen that DSFENet performed best for the details of buildings. The structure of the dual-stream branch balanced the deficiencies of the CNN and transformer, so that the extracted buildings retained the deficiencies of each single-branch structure. Compared with DSFENet-CNN with a single CNN branch, DSFENet combined with the transformer's global attention feature reduced some false detections and missed detections in the building detection, and the boundary details of the buildings were better. The results of the ablation experiments further demonstrated the validity of the combined effect of the two branches. A single transformer branch may perform poorly. When it is combined with a CNN branch, it can achieve a better performance than a single CNN branch. This makes it more suitable for building extraction.

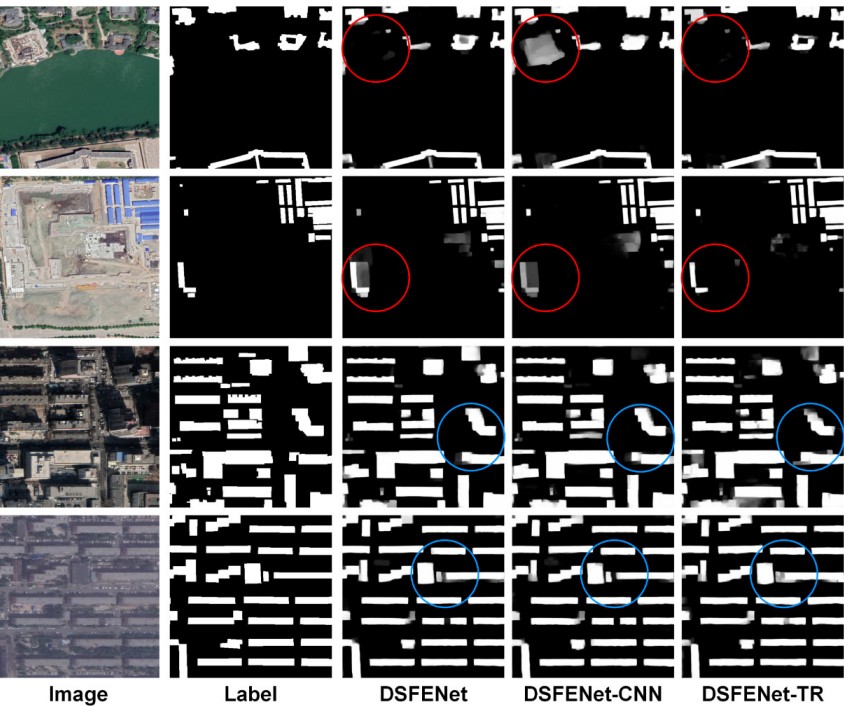

**Figure 10.** Comparison of DSFENet ablation experiments. The red circles illustrate that DSFENet-CNN had a certain false detection situation, while DSFENet-TR effectively alleviated this situation. The blue circles show that DSFENet-TR had poor prediction of building details, and DSFENet achieved a clearer building boundary prediction.

*4.6. Further Discussion*

From the perspective of global spatial relationships and local features of buildings, the aggregation of global and local information is meaningful. We proposed DSFENet based on a CNN and transformer. The encoder of DSFENet is composed of a CNN branch and a transformer branch. The convolution extracts the local features of buildings, and the self-attention mechanism achieves the global representation of buildings. DSFENet achieved accurate extraction of buildings.

With the development of a CNN architecture and transformer architecture, the combined design of the two has become the mainstream research direction. Moreover, due to the difference between the CNN features and transformer features, studying how to effectively combine CNN features and transformer features is meaningful. This paper proposes FAM, and experiments proved that the module is simple and effective. However, the FAM was designed without much consideration of the particularity of CNN features and transformer features. Therefore, the further design of a more effective combination method is our next goal.

The transformer architecture has the problems of a large parameter count, slow training speed, and slow prediction and reasoning speed. It is difficult to perform rapid applications in large-scale research area applications or real-time detection tasks. It is also difficult to apply it to some edge devices with insufficient computing power. Therefore, how to compress or redesign the model without decreasing the model accuracy is a challenging research direction. Models with fewer parameters facilitate fast learning and inference on data from this study area. This also makes deployment and detection in real time on edge devices possible, which is helpful for processing remote sensing information.

**5. Conclusions**

Considering the difference and complementarity between the feature modeling of the CNN model and transformer model, this paper proposed DSFENet, a neural network model combining a CNN branch and transformer branch as a dual-stream encoder. The CNN branch of DSFENet extracts local features, and the transformer branch of DSFENet performs global feature representation. The feature aggregation module performs the aggregation of features at each scale, to enhance the representation of features in the encoder. We found that there is a complementary relationship between low-level detail features and high-level semantic features. So we designed a semantic embedding module and a spatial embedding module. These could better combine low-level spatial information and high-level semantic information. According to the experimental results, the building extraction performance of DSFENet was better compared to the state-of-the-art models.

**Author Contributions:** L.X. and J.Z. designed and completed the experiments and wrote the article. Z.S., J.L., S.M. and Y.C. guided the process and helped with the writing of the paper. All authors have read and agreed to the published version of the manuscript.

**Funding:** This work was supported in part by the National Key Research and Development Program of China under Grant 2018YFB0505300, in part by the National Natural Science Foundation of China under Grant 41701472, Grant 42071316 and Grant 41971375.

**Data Availability Statement:** The code of DSFENet is publicly available at https://github.com/joyboy0209/DSFENet.

**Acknowledgments:** All authors sincerely thank the reviewers and editors for their helpful and detailed comments and suggestions.

**Conflicts of Interest:** The authors declare no conflict of interest.

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
