# Peer review of "Dual-Stream Feature Extraction Network Based on CNN and Transformer for Building Extraction"

_remotesensing, doi:10.3390/rs15102689_

Round 1

Reviewer 1 Report

The paper presents an approach for the segmentation of buildings in high resolution remote sensing images that is based on a deep neural network involving a CNN and Transformer branches for the extraction of features of various levels of abstraction. Two different fusion modules working on semantic and spatial embeddings are proposed as well. The proposed approach is shown to outperform the state-of-the-art on two benchmark datasets.

On the one hand, the proposed approach is scientifically sound and shows promising results against the state-of-the-art on both datasets. The conducted ablation studies seem to indicate the relevance of combining both convolutional and transformer-based feature extraction paths. Apart from an issue with the titles of the Sections (Section 2 should be named "Related Work" and all current titles shifted by one Section), I found the paper to be quite well written and easily understandable.

On the other hand, my concerns mostly concentrate on the novelty of the proposed approach. The association of a convolutional and transformer-based feature extraction branches for building segmentation seems to be a commonly applied idea in the related literature (a non-comprehensive list could include [1,2,3] for instance). I found Section 2.3 discussing related work to be somewhat shallow in particular. I would advise the authors to add more recent related work they may have missed (e.g. [1,2,3]). I also think it would be important to add a description of what differentiates their proposed approach from the ones from the related literature since this is currently missing.

I would finally advise the authors to carry out one last proofreading check to perform to eliminate some remaining typos (e.g. Figures 5 and 6).

References:

[1] L. Wang et al., UNetFormer: A UNet-like Transformer for Efficient Semantic Segmentation of Remote Sensing Urban Scene Imagery, ISPRS Journal of Photogrammetry and Remote Sensing, 2022

[2] X. He et al., Swin Transformer Embedding UNet for Remote Sensing Image Semantic Segmentation, IEEE Transactions on Geoscience and Remote Sensing, 2022

[3] C. Zhang et al., Transformer and CNN Hybrid Deep Neural Network for Semantic Segmentation of Very-High-Resolution Remote Sensing Imagery, IEEE Transactions on Geoscience and Remote Sensing, 2022

Author Response

Dear reviewer:

Thank you for your decision and constructive comments on my manuscript. We have carefully considered the suggestion of Reviewer and make some changes. We have tried our best to improve and made some changes in the manuscript.

The red part that has been revised according to your comments. Revision notes, point-to-point, are given as follows

Reviewer 2 Report

The authors have presented a dual-architecture of deep learning techniques, CNN and transformer, to segment building images based on the semantic and high-level features. The authors has shown promising direction of the usage of CNN and transformer as segmentation technique on aerial images.

The manuscript can be improved by addressing some comments as follows.

  1. Please elaborate on possible drawbacks of AGBEDNet, SSPDNet, and MAP-net as the CNN counterpart in Building Segmentation in Section 2.1.
  2. A little bit clear explanation might be needed for explaining whether [23] and [25] have high computation load and low edge classification accuracy in Section 2.2. 
  3. Please add details of the abbreviations of Figure 3: MLP, LN, W-MSA, and their importance as the part of transformer module.
  4. Section 2, Section 3, and Section 5 can be renamed into “Previous Research”, “Material and Methods”, and “Experimental Evaluation”, respectively, for better reading.
  5. Describing the Polygon Loss and Semantic Edge Loss and their purpose in Section 3.5 can add clarity to the manuscript. And please make the writing of “LOSS” to “Loss”.
  6. Please use the superscript to the learning rate parameter in Section 4.2.
  7. The evaluation metrics of P and R in section 4.3 can be described as Precision and Recall for more commonly used terms.
  8. The organization of the captions of Figures and Tables need to be adjusted to include them into the same page, not separated.

Author Response

(The authors gave the same response as above.)

Reviewer 3 Report

The paper titled "Dual-Stream Feature Extraction Network based on CNN and Transformer for Building Extraction" proposes a CNN-based and Transformer based Dual-Stream Feature Extraction Network (DSFENet) for accurate building extraction. As per the authors, convolution in the encoder extracts local features for buildings, and Transformer, during the encoding process, realizes global representation for buildings; therefore, a combination approach is used to obtain both local and global features. Though the approach looks promising, the following are major and minor changes: Major:  1. Both CNN (resnet-34) and Transformer (Swin) are merged for better feature extraction, what are the measures taken to ensure the lossless translation of transformer feature maps before merging? The functionality of the feature aggregation module is never clear and needs to be rewritten. 2. What is the purpose of the ASPP module introduced after the CNN branch and Swin branch? Why is ASPP used if the feature aggregation module is incorporated at every level? 3. What is the value addition by introducing SEEM and SPEM modules? From the article, the contribution of these modules is ambiguous. 4. Ablation study is briefly done; needs to be expanded as several modules are incorporated into the architecture.

Minor: 1. Diagram clarity and alignment issues are persistent. 2. Many statements are ambiguous and repetitive in several places. 3. Learning rate should be written as per the common practices. 4. Conclusion never mentions future directions of research.

Author Response

(The authors gave the same response as above.)

Round 2

Reviewer 1 Report

All my concerns have properly been addressed, so I recommend the acceptance of this paper.